# Prognostic Impacts of D816V *KIT* Mutation and Peri-Transplant *RUNX1–RUNX1T1* MRD Monitoring on Acute Myeloid Leukemia with *RUNX1–RUNX1T1*

**DOI:** 10.3390/cancers13020336

**Published:** 2021-01-18

**Authors:** Byung-Sik Cho, Gi-June Min, Sung-Soo Park, Silvia Park, Young-Woo Jeon, Seung-Hwan Shin, Seung-Ah Yahng, Jae-Ho Yoon, Sung-Eun Lee, Ki-Seong Eom, Yoo-Jin Kim, Seok Lee, Chang-Ki Min, Seok-Goo Cho, Dong-Wook Kim, Jong Wook-Lee, Myung-Shin Kim, Yong-Goo Kim, Hee-Je Kim

**Affiliations:** 1Department of Hematology, Catholic Hematology Hospital, Seoul St. Mary’s Hospital, College of Medicine, The Catholic University of Korea, Seoul 06591, Korea; cbscho@catholic.ac.kr (B.-S.C.); beichest@catholic.ac.kr (G.-J.M.); sspark@catholic.ac.kr (S.-S.P.); silvia.park@catholic.ac.kr (S.P.); royoon@catholic.ac.kr (J.-H.Y.); lee86@catholic.ac.kr (S.-E.L.); dreom@catholic.ac.kr (K.-S.E.); yoojink@catholic.ac.kr (Y.-J.K.); leeseok@catholic.ac.kr (S.L.); ckmin@catholic.ac.kr (C.-K.M.); chosg@catholic.ac.kr (S.-G.C.); dwkim@catholic.ac.kr (D.-W.K.); jwlee@catholic.ac.kr (J.-W.L.); 2Leukemia Research Institute, College of Medicine, The Catholic University of Korea, Seoul 06591, Korea; 3Department of Hematology, Yeouido St. Mary’s Hospital, College of Medicine, The Catholic University of Korea, Seoul 06591, Korea; native47@catholic.ac.kr; 4Department of Hematology, Eunpyeong St. Mary’s Hospital, College of Medicine, The Catholic University of Korea, Seoul 06591, Korea; chironhmt@catholic.ac.kr; 5Department of Hematology, Incheon St. Mary’s Hospital, College of Medicine, The Catholic University of Korea, Seoul 06591, Korea; saymd@catholic.ac.kr; 6Department of Laboratory Medicine, Seoul St. Mary’s Hospital, College of Medicine, The Catholic University of Korea, Seoul 06591, Korea; microkim@catholic.ac.kr (M.-S.K.); yonggoo@catholic.ac.kr (Y.-G.K.)

**Keywords:** AML, *RUNX1–RUNX1T1*, D816V *KIT* mutation, hematopoietic stem cell transplantation, measurable residual disease

## Abstract

**Simple Summary:**

Acute myeloid leukemia (AML) with *RUNX1-RUNX1T1* is a heterogeneous disease entailing different prognoses. Patients with high-risk features can benefit from allogeneic hematopoietic stem cell transplantation (HSCT) or autologous HSCT. However, insufficient data about major risk factors, such as *KIT* mutations and measurable residual disease (MRD) status for relapse, make it difficult to clarify the benefit of each transplant strategy. Moreover, limited data are available to elucidate the exact prognostic impacts of different types of *KIT* mutations and optimal thresholds or time points for *RUNX1–RUNX1T1* MRD assessment, particularly in the setting of HSCT. Given the lack of prospective study, the current retrospective study, including a large cohort of high-risk AML patients with *RUNX1–RUNX1T1*, firstly demonstrated the differentiated prognostic impact of D816V *KIT* mutation among various *KIT* mutations and clarified optimal time points and thresholds for *RUNX1–RUNX1T1* MRD monitoring in the setting of HSCT.

**Abstract:**

The prognostic significance of *KIT* mutations and optimal thresholds and time points of measurable residual disease (MRD) monitoring for acute myeloid leukemia (AML) with *RUNX1-RUNX1T1* remain controversial in the setting of hematopoietic stem cell transplantation (HSCT). We retrospectively evaluated 166 high-risk patients who underwent allogeneic (Allo-HSCT, *n* = 112) or autologous HSCT (Auto-HSCT, *n* = 54). D816V *KIT* mutation, a subtype of exon 17 mutations, was significantly associated with post-transplant relapse and poor survival, while other types of mutations in exons 17 and 8 were not associated with post-transplant relapse. Pre- and post-transplant *RUNX1–RUNX1T1* MRD assessments were useful for predicting post-transplant relapse and poor survival with a higher sensitivity at later time points. Survival analysis for each stratified group by D816V *KIT* mutation and pre-transplant *RUNX1–RUNX1T1* MRD status demonstrated that Auto-HSCT was superior to Allo-HSCT in MRD-negative patients without D816V *KIT* mutation, while Allo-HSCT was superior to Auto-HSCT in MRD-negative patients with D816V *KIT* mutation. Very poor outcomes of pre-transplant MRD-positive patients with D816V *KIT* mutation suggested that this group should be treated in clinical trials. Risk stratification by both D816V *KIT* mutation and *RUNX1–RUNX1T1* MRD status will provide a platform for decision-making or risk-adapted therapeutic approaches.

## 1. Introduction

Acute myeloid leukemia (AML) with *RUNX1-RUNX1T1* is known to have a favorable prognosis. However, it is also a heterogeneous disease entailing different prognoses [1,2,3]. Several prognostic factors, including *KIT* mutations and measurable residual disease (MRD) status, have been proposed [1,2,3,4,5,6,7,8,9,10,11]. Many types of *KIT* mutations have been identified in AML, and their prognostic significance has been conflicting so far [12]. Thus, the current European Leukemia Net (ELN) guideline does not support the use of *KIT* mutational status in clinical guidance in terms of therapeutic intervention [13]. However, recent studies have suggested different prognostic significance of each type of *KIT* mutations [8,9,14]. A recent prospective study has demonstrated that *KIT* mutations in exon 17 among three mutation hot-spots (exon 8, exon 10–11, and exon 17) are only prognostic for AML with *RUNX1-RUNX1T1* through evaluation of all types of *KIT* mutations [8]. *RUNX1-RUNX1T1* quantification by real-time quantitative polymerase chain reaction (RT-qPCR) is useful as an MRD tool for predicting relapse [15]. Some groups have suggested that *RUNX1-RUNX1T1* quantification has even better predictability than *KIT* mutations [1,10,16]. However, there are several opinions regarding thresholds and time points for MRD assessment with *RUNX1-RUNX1T1* [1,10,15,16,17,18]. Those optimal thresholds and time points should be evaluated according to types of *KIT* mutations and post-remission therapy.

Allogeneic hematopoietic stem cell transplantation (Allo-HSCT) can benefit high-risk AML patients with *RUNX1-RUNX1T1* [18], although it is not generally recommended during the first complete remission (CR) [13]. A few previous reports have demonstrated that autologous hematopoietic stem cell transplantation (Auto-HSCT) had similar survival rates with Allo-HSCT [19,20,21,22,23]. However, those comparisons were limited by insufficient data about *KIT* mutations and MRD status to clarify the benefit of each transplant strategy. Therefore, we evaluated prognostic significance of different types of *KIT* mutations and *RUNX1-RUNX1T1* quantification in high-risk AML patients with *RUNX1-RUNX1T1* who underwent Allo-HSCT or Auto-HSCT to clarify the clinical relevance of each transplant strategy in each risk group. Furthermore, we elucidated optimal thresholds and time points of *RUNX1-RUNX1T1* quantification during the peri-transplant period.

## 2. Results

### 2.1. Patient Characteristics and KIT Mutations

A total of 166 patients with a median age of 40 years (range, 18–69 years) underwent Allo- or Auto-HSCT in the first CR (CR1, *n* = 156) and second CR (CR2, *n* = 10). The patient-, disease-, and transplant-related characteristics according to *KIT* mutations are summarized in Table 1. *KIT* mutations were detected in 70 (42%) of 166 patients. Among *KIT* exon 17 mutations, D816V mutation was identified the most frequently, followed by N822K, D816H, D816Y, and exon 8 mutations. Eighteen (26%) of the *KIT*-mutated patients had multiple *KIT* mutations. There was no significant difference in patient- or disease-related characteristics between *KIT-*unmutated and *KIT*-mutated patients. In patients with *KIT* mutations, Allo-HSCT was more frequently performed than Auto-HSCT, while proportions of Allo-HSCT and Auto-HSCT were similar in patients without *KIT* mutations, resulting in differences of variables related to transplant procedures, including donor types, stem cell source, and transplanted CD34^+^ cell number.

### 2.2. Impact of KIT Mutations Status on RUNX1–RUNX1T1 MRD Kinetics and Survival Outcomes

Impacts of *KIT* mutations on *RUNX1–RUNX1T1* MRD kinetics were evaluated (Table 2). *RUNX1–RUNX1T1* levels were continuously decreased after chemotherapies and transplantation. Mean bone marrow (BM) log_10_-transformed transcript levels at post-induction, pre-HSCT, and 1 month after HSCT were significantly higher and degrees of log reduction were significantly lower in *KIT*-mutated patients than *KIT*-unmutated patients. Patients with D816V *KIT* mutation had significantly less reduction in *RUNX1–RUNX1T1* level compared to patients without D816V *KIT* mutation. For the other hot mutations, including D816Y, D816H, N822K, or exon 8 mutations, there was no significant difference in *RUNX1–RUNX1T1* MRD kinetics between patients with and without such mutations (data not shown). These data suggest that *KIT* mutations, particularly D816V *KIT* mutation, have some resistance to chemotherapies and/or transplantation. 

At a median follow-up of 60 months (range, 6–131 months), 21 patients relapsed at a median of 8 months (range, 4–25 months) after HSCT. A higher trend of cumulative incidence of relapse (CIR) in *KIT*-mutated patients was observed (Figure 1A and Appendix A). Multivariate analysis with an adjustment relating to disease status at transplant revealed significant associations between *KIT* mutations and CIR (Appendix A). However, a significant difference of CIR was only observed in the group with Auto-HSCT, whereas there was no significant difference of CIR in the group with Allo-HSCT (Figure 1B,C). In a subgroup analysis according to types of *KIT* mutations, patients with D816V *KIT* mutation had significantly increased CIR compared to patients with other *KIT* mutations or without *KIT* mutations, while other types of mutations in exon 17 or 8 were not associated with post-transplant relapse (Appendix A). Moreover, patients with D816V *KIT* mutation had significantly higher CIR in both Allo-HSCT and Auto-HSCT groups than patients without such mutation (Figure 1D–F), which translated into inferior disease-free survival (DFS) and overall survival (OS, Appendix A).

### 2.3. Optimal Time Points and Thresholds for RUNX1–RUNX1T1 MRD Monitoring

Results of serial RT-qPCR assays of *RUNX1–RUNX1T1* according to transplant type are presented in Appendix A. Both Allo-HSCT and Auto-HSCT significantly decreased *RUNX1–RUNX1T1* levels after transplant (Figure 2A). *RUNX1–RUNX1T1* levels at pre-HSCT were significantly higher in the group of Allo-HSCT, which had more D816V *KIT*-mutated patients, than in the group of Auto-HSCT. Such difference persisted at 1 month after transplantation. Their levels became similar at 3 months after transplantation (Figure 2A and Appendix A). Relapsed patients had significantly higher *RUNX1–RUNX1T1* levels at pre-HSCT and at 1 or 3 months after HSCT than non-relapsed patients. Such levels in relapsed patients were not significantly decreased after HSCT compared to the progressive decrease in non-relapsed patients (Figure 2B). Receiver operating characteristic (ROC) curve analysis revealed that *RUNX1–RUNX1T1* levels at all time points, including pre-HSCT and 1 or 3 months after HSCT, could predict post-transplant relapse (Figure 2C). The sensitivity and specificity at each time point (Appendix A) showed a trend of improvement as time went by. 

To elucidate optimal thresholds of RUNX1–RUNX1T1 levels for predicting post-transplant relapse, various cutoffs, including copy numbers (1000 copies, 500 copies, 250 copies, 100 copies, 50 copies, 10 copies, and 0 copy) and degrees of log reduction (3 log or 4 log) at each time point, were compared (Appendix A). Every cutoff level was useful for identifying patients at high risk or relapse. A 3 log reduction at each time point appeared to be the most effective one based on sensitivity and specificity (pre-HSCT, 38% and 91%; 1 month after HSCT, 75% and 89%; 3 months after HSCT, 83% and 94%; Appendix A). Multivariate analysis of significant factors found in univariate analysis (Appendix A) revealed that MRD positivity defined by 3 log reduction at each time point independently predicted CIR, which translated into inferior DFS and OS (Appendix A). Figure 3 and Appendix A show survival outcomes according to MRD positivity defined by 3 log reduction.

### 2.4. Prognostic Independency of D816V KIT Mutation and RUNX1–RUNX1T1 MRD Status

Multivariate analysis was performed to compare prognostic significance for predicting post-transplant relapse among three major factors found in univariate analysis (Appendix A): disease status at transplant, D816V *KIT* mutation, and *RUNX1–RUNX1T1* MRD status defined by 3 log reduction (Model #2 in Table 3). It revealed that MRD status at each time point and disease status at transplant were significant for predicting post-transplant relapse. D816V *KIT* mutation remained significant in the multivariate model. We also evaluated prognostic independency of *KIT* mutations shown in Appendix A and *RUNX1–RUNX1T1* MRD status (Model #1 in Table 3). In contrast to D816V *KIT* mutation, *KIT* mutations including all types lost their significance in the multivariate model.

### 2.5. Outcomes of Transplant Type in Each Group Stratified by D816V KIT Mutation and RUNX1–RUNX1T1 MRD Status at Pre-HSCT

Univariate analysis showed that the Auto-HSCT group had favorable DFS and OS due to both reduced non-relapse mortality (NRM) rates and similar CIR compared to Allo-HSCT group (Appendix A). However, multivariate models (Appendix A) revealed no significant difference in DFS or OS between Allo- and Auto-HSCT groups, which might be due to significant differences in patient- and/or disease-related characteristics such as older age, more CR2, and higher number of white blood cell (WBC) counts at diagnosis and transplanted CD34+ cells in the Allo-HSCT group than Auto-HSCT group (Appendix A). In particular, proportions of D816V *KIT* mutation and *RUNX1–RUNX1T1* levels at pre-HSCT, which were demonstrated as important factors associated with post-transplant relapse, were significantly higher in the Allo-HSCT group than in the Auto-HSCT group.

Thus, we stratified patients by D816V *KIT* mutation and MRD positivity defined by 3 log reduction in *RUNX1–RUNX1T1* at pre-HSCT into four groups to elucidate effects of transplant type in each risk group. Patients with both D816V *KIT* mutation and MRD positivity had significantly increased CIR compared to those in the other three groups (Figure 4A). In subgroup analyses for effects of transplant type in each group (Figure 4B–E and Appendix A), among MRD-negative patients, Allo-HSCT was more beneficial to prevent relapse in patients with D816V *KIT* mutation than Auto-HSCT, resulting in superior DFS and OS for patients in the Allo-HSCT group. In contrast, superior DFS and OS of Auto-HSCT to Allo-HSCT were observed in patients without D816V *KIT* mutation due to improved NRM in Auto-HSCT without significant difference in CIR. Among MRD-positive patients, effects of transplant type were not clear due to the small number of patients in each group. However, we observed that patients with D816V *KIT* mutation had significantly higher risk for relapse than patients without such mutation, which translated into very poor DFS and OS. NRM rather than CIR was a determinant factor for survival in MRD-positive patients without D816V *KIT* mutation who underwent Allo-HSCT.

## 3. Discussion

Allo-HSCT is not generally considered during CR1 for AML with *RUNX1–RUNX1T1* [13]. However, patients with high-risk features can benefit from Allo-HSCT [18], and recent data suggest that Auto-HSCT may be an alternative [19,20,21,22,23]. Nevertheless, insufficient data about major risk factors, such as *KIT* mutations and MRD status for relapse, make it difficult to clarify the benefit of each transplant strategy. Moreover, limited data are available to elucidate prognostic impacts of different types of *KIT* mutations and optimal thresholds or time points for *RUNX1–RUNX1T1* MRD assessment, particularly in the setting of HSCT. The current study, including high-risk AML patients with *RUNX1–RUNX1T1* who underwent Allo- and Auto-HSCT. demonstrated that both D816V *KIT* mutation and peri-transplant *RUNX1–RUNX1T1* MRD monitoring were independently associated with post-transplant relapse and survival. Of note, in analyses of stratified groups according to the presence of D816V *KIT* mutation and pre-transplant MRD status, Allo-HSCT and Auto-HSCT were superior to each other in MRD-negative patients with and without D816V *KIT* mutation, respectively. In addition, poor outcomes in pre-transplant MRD-positive patients with D816V *KIT* mutation due to increased post-transplant relapse suggest the necessity of pre- or post-transplant therapeutic targeting with small molecules against the *RUNX1–RUNX1T1* protein, the use of tyrosine kinase inhibitors (dasatinib and FLT3 inhibitors), epigenetic modulators [24], and cellular approaches [25,26] for this group of patients.

Prognostic impacts of *KIT* mutations in core-binding factor AML, including *RUNX1–RUNX1T1* and *CBFB-MYH11*, have been controversial [12]. Thus, they are not included in guidelines for risk stratification in the current ELN guideline [13]. However, recent studies have reported the prognostic importance of *KIT* mutations in AML with *RUNX1–RUNX1T1,* but not in those with *CBFB-MYH11* [8,23]. Prognostic impacts of each type of *KIT* mutations may be different [8,9,14]. Compared to other *KIT* mutations, exons 17 and/or 8 mutations are more likely to adversely affect survival [8,9,14]. The German–Austrian AML Study Group (AMLSG) has demonstrated correlations of *RUNX1–RUNX1T1* levels after chemotherapies with *KIT* mutations (exons 17 and/or 8), with *KIT*-unmutated patients achieving deeper MRD reductions at the end of treatment [10]. This indicates that *KIT*-mutated patients have some resistance to chemotherapies. A recent prospective study including patients treated with chemotherapy only has demonstrated that *KIT* mutations in exon 17 are prognostic factor in AML with *RUNX1/RUNX1T1* after evaluating all types of *KIT* mutations [8]. In our unique cohort of transplanted high-risk patients, we firstly showed that D816V *KIT* mutation, a subtype of exon 17 mutations, rather than all *KIT* mutations, was significantly associated post-transplant relapse and poor survival. Indeed, other types of mutations in exons 17 (D816H, D816Y, or N822K) and 8 showed no significant association with increased risk for relapse. Less reduction in *RUNX1–RUNX1T1* level in D816V *KIT-*mutated patients after not only chemotherapy, in line with AMLSG, but also HSCT, supports poor prognostic features related with relapse. A recent study by Tarlock et al. investigated the functional impact of distinct mutation subsets of *KIT* mutations in an in vitro model [27]. They demonstrated that the D816V mutation resulted in more potent KIT phosphorylation as well as increased detection of immature vs. mature form of KIT compared with N822K or exon 8 mutations, which supports the distinctive prognostic role of D816V *KIT* mutation in our study. Some reports have suggested that the prognostic impacts of *KIT* mutations were outweighed by *RUNX1–RUNX1T1* level during treatment in the setting of chemotherapy [1,10] or HSCT [16]. In contrast, our results showed that D816V *KIT* mutation remained a significant factor for post-transplant relapse with *RUNX1–RUNX1T1* MRD status in multivariate models, suggesting its independent prognostic power, which needs to be further evaluated in a larger cohort. The prognostic importance of *KIT* mutation in specific loci in the current study, as well as in other recent reports [8,9,14], provides evidences that the current ELN guidelines need to be revised to include *KIT* mutational status for risk stratification.

Several studies have demonstrated that the persistent presence of *RUNX1–RUNX1T1* transcript is a strong predictor of relapse, while optimal time points or thresholds during the active treatment phase remain controversial [15]. Recently, AMLSG has proposed a refined practical guidance for *RUNX1–RUNX1T1* MRD monitoring, emphasizing the achievement of MRD negativity in both BM and PB after completion of therapy [10], in contrast to suggested importance of >3 log reduction in BM between diagnosis and the end of induction [13] or consolidation [28] in previous studies. However, few data are available about whether *RUNX1–RUNX1T1* MRD monitoring can continue to serve as an efficient tool for risk stratification after HSCT. An earlier study from China has suggested that not pre-transplant but post-transplant *RUNX1–RUNX1T1* MRD monitoring with a cutoff of 3 log reduction could discriminate patients at high risk of post-transplant relapse, and post-transplant *RUNX1–RUNX1T1* MRD monitoring was also more predictive of relapse risk than *KIT* mutations [16]. In contrast, our data with a larger cohort revealed that pre- and post-transplant *RUNX1–RUNX1T1* MRD monitoring were useful for predicting post-transplant relapse. They were more predictive than whole *KIT* mutations, similar to data from China. However, they had similar predictive power to the D816V *KIT* mutation. Furthermore, based on sensitivity and specificity at pre-HSCT and at 1 or 3 months after HSCT, our results clearly demonstrated that the optimal threshold was not an achievement of MRD negativity, but 3 log reductions in the setting of HSCT. The sensitivity was greater at 3 months after HSCT, whereas the specificity was similar at all time points. Thus, the optimal time point for *RUNX1–RUNX1T1* MRD monitoring might be 3 months after HSCT, which would be available to apply additional therapies in an attempt to prevent relapse because all relapse events were observed beyond 4 months after HSCT. On the other hand, earlier recognition of high-risk patients at pre-HSCT would be helpful for applying additional cellular approaches to enhance anti-leukemia effects, which needs considerable time for preparation [25,26]. Further studies are warranted to validate the role of risk-adapted approaches based on *RUNX1–RUNX1T1* MRD monitoring as well as *KIT* mutations.

The current study evaluated a large cohort of high-risk AML patients with *RUNX1–RUNX1T1* who underwent Allo-HSCT or Auto-HSCT to clarify the prognostic significance of different types of *KIT* mutations and *RUNX1/RUNX1T1* quantification. A limitation of our study was that we focused on *KIT* mutations of exons 17 and 8. These mutations have been reported to adversely affect survival of AML patients with *RUNX1–RUNX1T1* [8,9,14]. Interactions of *KIT* mutations with other concurrent mutations, such as *ASXL1* [14] or the recently identified *SMC1A* and *DHX15* [29], need to be further evaluated. Differences in pre-transplant characteristics, particularly *RUNX1/RUNX1T1* level and presence of *KIT* mutations, between Allo- and Auto-HSCT due to the retrospective nature of this study and transplant decisions based on donor availability should be considered when interpreting our data. That is why we stratified all patients into four groups based on both *RUNX1/RUNX1T1* levels and D816V *KIT* mutation, and compared Allo-HSCT with Auto-HSCT in each group. Different regimens for induction, including gemtuzumab ozogamicin in patients with CD33 or dasatinib in patients with *KIT* mutations, or consolidation, such as high-dose cytarabine, may affect *RUNX1–RUNX1T1* MRD levels and outcomes [30]. Thus, the combined impact of *KIT* mutations and *RUNX1–RUNX1T1* MRD status should be further evaluated in patients treated with those regimens without transplantation.

## 4. Materials and Methods

### 4.1. Patients

We retrospectively evaluated 183 consecutive AML patients with *RUNX1/RUNX1T1* in remission who underwent HSCT at the Catholic Hematology Hospital between 2009 and 2018. After excluding nine patients who underwent second HSCT and eight patients without data of *KIT* mutations (*n* = 3) or pre-transplant *RUNX1/RUNX1T1* MRD data (*n* = 5), 166 patients were finally included in this study (Appendix A). Before transplant, 141 of 166 patients were classified as high-risk AML by the persistence of *RUNX1/RUNX1T1* transcript (*n* = 130, 78%) and/or *KIT* mutations (*n* = 70, 42%). Twenty patients had other high-risk features such as loss of Y chromosome (*n* = 17) [3,31], second CR (*n* = 2), and extramedullary disease (*n* = 1) [32], while five patients persisted to undergo HSCT. All patients were treated with intensive induction and one or two cycles of consolidation chemotherapy [33]. Patients underwent Allo-HSCT if an available donor was found during consolidation. If the patient did not have an available donor, patients underwent Auto-HSCT. For Auto-HSCT, CD34^+^ stem cells were collected for three days after neutrophil count recovered from consolidation chemotherapy [34]. The Institutional Review Board of the Catholic Medical Center approved the current study (#KC16TISI0438). All analyses were performed according to the Institutional Review Board guidelines and tenets of the Declaration of Helsinki.

### 4.2. Chemotherapy and Transplant Procedures

All patients were treated according to our standard protocol, consisting of “3 + 7” idarubicin (IDA, 12 mg/m^2^, intravenous infusion) plus cytarabine (Ara-C, 100 mg/m^2^ continuously infused for 24 h) for remission-induction chemotherapy. After achieving CR, two consolidation chemotherapies were administered. Consolidation chemotherapies consisted of “3 + 5” mitoxantrone (12 mg/m^2^, intravenous infusion) or IDA (12 mg/m^2^, intravenous infusion) plus an intermediate dose of Ara-C (1.0 g/m^2^, intravenous infusion, bid), which were alternated. Transplant-related characteristics of the patients according to transplant type are listed in Appendix A. For transplants from matched sibling or unrelated donors, myeloablative or reduced-intensity conditioning regimens consisting of fludarabine (150 mg/m^2^) and busulfex (6.4 mg/kg), with or without fractionated total body irradiation (TBI) of 400 cGy, were used based on age and/or comorbidity [35]. For transplants from haploidentical donors, all patients received a reduced-intensity toxicity conditioning regimen consisting of fludarabine (150 mg/m^2^), busulfex (6.4 mg/kg), and fractionated TBI (800 cGy) [35]. For Auto-HSCT, an myeloablative regimen consisting of TBI (1200 cGy), cytarabine (9 g/m^2^), and melphalan (100 mg/m^2^) was used for the majority (91%) of patients [35]. Graft-versus-host disease prophylaxis was performed using a short-course methotrexate plus cyclosporine for transplants from matched sibling donors or tacrolimus for transplants from matched unrelated and haploidentical donors. Anti-thymocyte globulin (Sanofi/Genzyme, Cambridge, MA, USA) was given at a fixed dose of 5.0 mg/kg for transplants from haploidentical donors, while different doses (1.25–5.0 mg/kg) were used over different time periods for transplants from matched unrelated donors. Other general transplant-related procedures were performed as previously described [34].

### 4.3. Cytogenetic and Molecular Analyses

For detection of karyotypes, BM was used. At least 20 metaphases were analyzed with the Giemsa banding method after 24 or 48 h of unsynchronized culture. The presence of RUNX1/RUNX1T1 was confirmed by a multiplex reverse transcriptase polymerase chain reaction screening assay using a HemaVision Kit (DNA Technology, Risskov, Denmark). Mutations in exons 8 and 17 of the KIT gene were analyzed as previously reported [35]. Screening tests for detecting KIT mutations (D816V, D816H, D816Y and N822K) in exon 17 were performed with an allele-specific RT-qPCR assay (Real-Q KIT Screening Kit; BioSewoom, Seoul, Korea). Positive samples were genotyped using a Real-Q KIT Genotyping Kit (BioSewoom) for discriminating each mutation. Detection of KIT mutations in exon 8 was performed by direct sequencing [36]. Mutations in FLT3, NPM1, and CEBPA were also analyzed using protocols established since 2008 [34,36].

### 4.4. RUNX1–RUNX1T1 MRD Assessment and Definitions

For the assessment of MRD, BM samples at pre-HSCT and at 1 or 3 months after HSCT were used for RT-qPCR for *RUNX1–RUNX1T1*. Total RNA was isolated from patients’ BM aspirates using the High Pure RNA Isolation Kit (Roche Diagnostics, Mannheim, Germany). Nucleic acid quality and quantity was measured using a NanoDrop 1000 spectrophotometer (Thermo Fisher Scientific, Waltham, MA, USA). Reverse transcription was carried out using Transcriptor First Strand cDNA Synthesis Kit (Roche Applied Science, Mannheim, Germany). RT-qPCR assay was performed using the Real-Q™ *RUNX1–RUNX1T1* Quantification kit (Bioseum, Seoul, Korea) according to the manufacturer’s instruction. The kit consisted of *RUNX1–RUNX1T1* and *ABL1* standard materials of four for each concentration. After preparing PCR mixture for *RUNX1–RUNX1T1* and *ABL1* (4 µL of each probe and primer mixture, 4 µL of cDNA or standard material, 12.5 µL of PCR reaction mixture, and 4.5 µL of distilled water), reactions were performed using an ABI 7500 Real-Time PCR system (Applied Biosystems, Foster City, CA, USA). PCR conditions were 2 min at 50 °C, 10 min at 95 °C, 45 cycles of 15 s at 95 °C and 1 min at 60 °C. *RUNX1–RUNX1T1* fusion genes calculated with standard materials were normalized with respect to the number of *ABL1* transcripts and expressed as copy numbers per 1 × 10^5^ copy of *ABL1*. Assays were performed in replicate with appropriate controls. We tested different thresholds for levels of *RUNX1–RUNX1T1* or log reduction compared to levels at diagnosis. ROC curve analysis was used to determinate optimal time points for MRD assessment with *RUNX1–RUNX1T1.* Please refer to Appendix A for detailed descriptions for quality control of the MRD assessment.

### 4.5. Statistical Analysis

The categorical variables were compared using the Chi-square test or Fisher’s exact test while the continuous variables were analyzed with the Student’s t-test or Wilcoxon’s rank-sum test. OS and DFS curves were plotted using the Kaplan–Meier method and analyzed with the log-rank test. The cumulative incidence was used to estimate the probability of the CIR and NRM. Non-relapse death and relapse were treated as competing risk factors for CIR and NRM, respectively, and compared using the Gray test. Multivariate analysis included variables with *p*-values of <0.10, as determined by univariate analysis, and were considered for entry into the model selection procedure based on the Cox proportional hazards model or a proportional hazards model for the sub-distribution of the competing risk factors. Statistical significance was indicated by a *p*-value of less than or equal to 0.05 (two-tailed). All statistical analyses were conducted using SPSS, version 13.0 (SPSS, Inc., Chicago, IL, USA) and R-software (version 3.4.1, R Foundation for Statistical Computing, 2017).

## 5. Conclusions

This study demonstrated the differentiated prognostic impact of D816V *KIT* mutation among various *KIT* mutations and clarified optimal time points and thresholds for *RUNX1–RUNX1T1* MRD monitoring in the setting of HSCT. Both D816V *KIT* mutation and peri-transplant MRD monitoring with *RUNX1–RUNX1T1* were independently associated with post-transplant relapse and survival. This study also revealed the relevance of each transplant type in each risk group by favorable outcomes of Auto-HSCT in MRD-negative patients without D816V *KIT* mutation and Allo-HSCT in patients with MRD positivity or D816V *KIT* mutation. Effects of transplant type in MRD-positive patients need to be further evaluated in a larger cohort. However, very poor outcomes in pre-transplant MRD-positive patients with D816V *KIT* suggested that this group should be treated in clinical trial and research. Finally, our data suggested that risk stratification by both D816V *KIT* mutation and *RUNX1–RUNX1T1* MRD status could provide a platform for decision-making or risk-adapted therapeutic approaches.

## Figures and Tables

**Figure 1 cancers-13-00336-f001:**
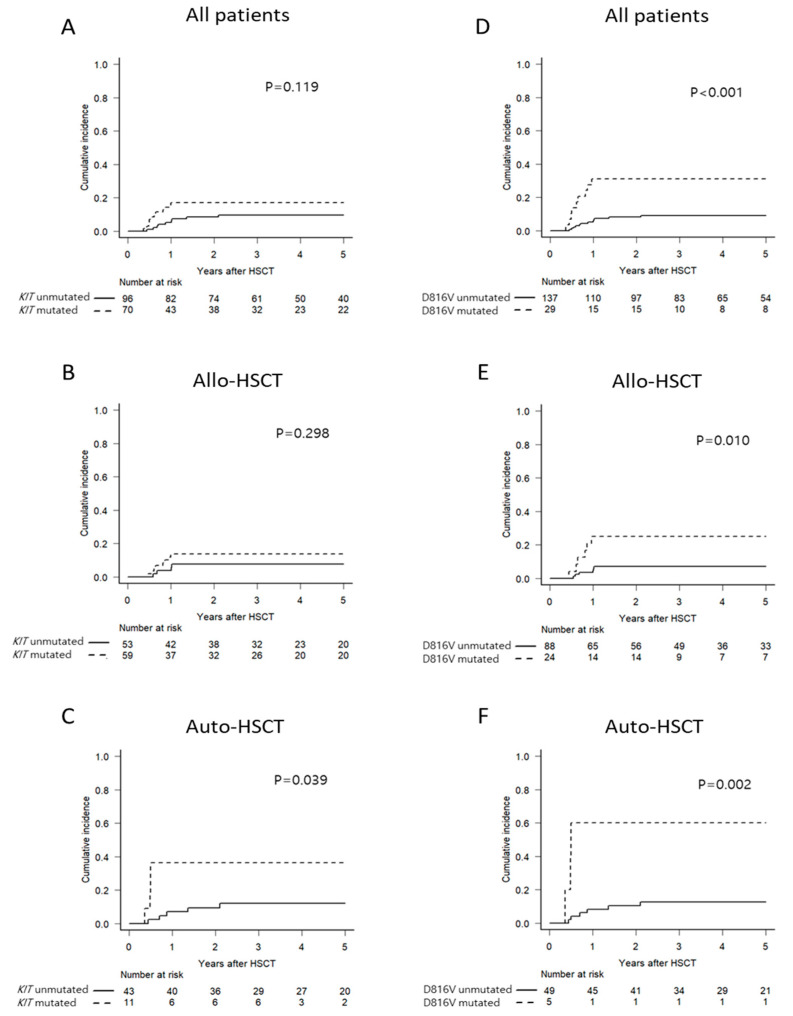
Cumulative incidence of relapse according to *KIT* mutations (**A**–**C**) or D816V *KIT* mutation (**D**–**F**) in all patients (**A**,**D**) or patients in Allo-HSCT (**B**,**E**) and Auto-HSCT (**C**,**F**) groups.

**Figure 2 cancers-13-00336-f002:**
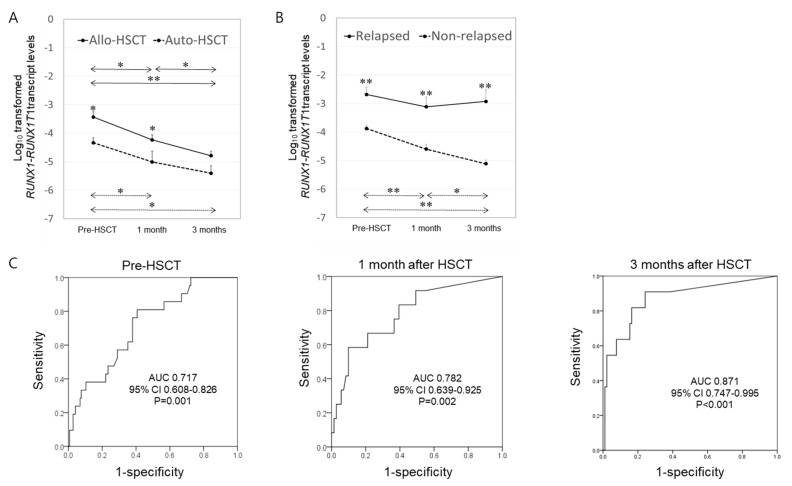
*RUNX1–RUNX1T1* expression kinetics during peri-transplant period and optimal time points. (**A**) *RUNX1–RUNX1T1* expression kinetics according to transplant type (Allo-HSCT vs. Auto-HSCT) before HSCT and at 1 or 3 months after HSCT (numbers of Allo-HSCT vs. Auto-HSCT; before, 112 vs. 54; 1 month, 70 vs. 13; 3 months, 86 vs. 16). (**B**) *RUNX1–RUNX1T1* expression kinetics according to the occurrence of relapse (numbers of relapsed patients vs. non-relapsed patients; before, 21 vs. 145; 1 month, 12 vs. 71; 3 months, 11 vs. 91). (**C**) Receiver operating characteristic (ROC) curve analysis of *RUNX1–RUNX1T1* levels at each time point. *RUNX1–RUNX1T1* levels were normalized to the number of *ABL1* transcripts and expressed as copy numbers per 10^5^ copies of *ABL1.* Results are expressed as mean ± SEM. * *p* < 0.05, ** *p* < 0.01. AUC indicates area under curve.

**Figure 3 cancers-13-00336-f003:**
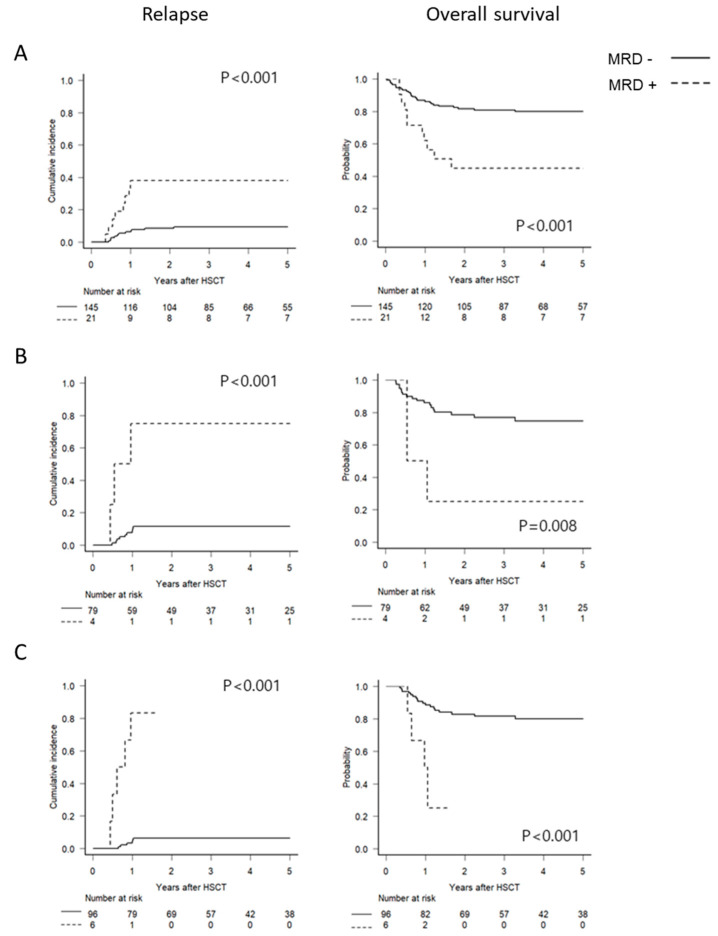
Survival outcomes according to MRD positivity defined by 3 log reduction in *RUNX1–RUNX1T1* levels at pre-HSCT (**A**) and at 1 month (**B**) or 3 months (**C**) after HSCT.

**Figure 4 cancers-13-00336-f004:**
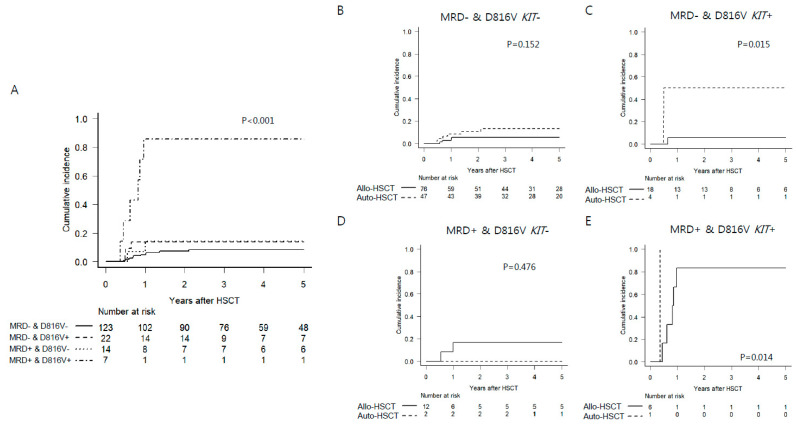
Cumulative incidence of relapse (CIR) according to both D816V *KIT* mutation and MRD positivity defined by 3 log reduction in *RUNX1–RUNX1T1* levels at pre-HSCT (**A**) and effects of transplant types on CIR in each stratified group (**B**–**E**).

**Table 1 cancers-13-00336-t001:** Patient-, disease-, and transplant-related characteristics according to KIT mutations.

Variables	Overall (*n* = 166)	*KIT* Unmutated (*n* = 96)	*KIT* Mutated (*n* = 70)	*p*
Age at transplantation, years				
Median (range)	40 (18–69)	38 (18–64)	42 (18–69)	0.529
Sex, *n* (%)				0.814
Male	105 (63)	60 (64)	45 (64)	
Female	61 (37)	36 (36)	25 (36)	
AML type, *n* (%)				0.074
De novo	161 (97)	91 (95)	70 (100)	
Therapy-related	5 (3)	5 (5)	0	
WBC count per liter at diagnosis				
Median (range)	8.65 (0.53–100.91)	7.19 (1.33–100.91)	10.80 (0.53–68.6)	0.206
Additional cytogenetic abnormalities, *n* (%)				
Del(9q)	12 (7)	5 (5)	7 (10)	0.239
Trisomy 8	2 (1)	1 (1)	1 (1)	1.000
Loss of sex chromosome	104 (63)	64 (67)	40 (57)	0.210
Del(7q)	3 (2)	1 (1)	2 (3)	0.574
Complex karyotype	9 (5)	4 (4)	5 (7)	0.495
*KIT* mutations *n* (%)				
Exon 17-D816V	29 (18)	0	29 (41)	-
Exon 17-D816Y	14 (8)	0	14 (20)	-
Exon 17-D816H	19 (11)	0	19 (27)	-
Exon 17-N822K	25 (15)	0	25 (36)	-
Exon 8	5 (3)	0	5 (7)	-
*FLT3* mutations, *n* (%)				
* FLT3-ITD*	9 (5)	6 (6)	3 (4)	0.379
* FLT3-TKD*	3 (2)	1 (1)	2 (3)	0.327
Missing data	9 (5)	7 (7)	2 (3)	
Disease status at HSCT, *n* (%)				0.194
CR1	156 (94)	88 (92)	68 (97)	
CR2	10 (6)	8 (8)	2 (3)	
Donor type, *n* (%)				0.001
Matched sibling	64 (39)	34 (35)	30 (43)	
Matched unrelated	25 (15)	10 (10)	15 (21)	
Haploidentical	23 (14)	9 (9)	14 (20)	
Autologous	54 (32)	43 (45)	11 (16)	
Stem cell source, *n* (%)				0.003
Peripheral blood	111 (67)	55 (57)	56 (80)	
Bone marrow	28 (17)	18 (19)	10 (14)	
Peripheral blood and bone marrow	27 (16)	23 (24)	4 (6)	
Conditioning intensity, *n* (%)				0.256
Myeloablative	105 (63)	57 (59)	48 (69)	
Reduced intensity	61 (37)	39 (41)	22 (31)	
Interval from diagnosis to transplant, days				
Median (range)	194 (96–260)	195 (96–260)	184 (102–243)	0.174
CD34^+^ cells × 10^6^/kg in graft				
Median (range)	3.88 (0.73–16.73)	3.52 (1.01–16.10)	4.88 (0.73–16.73)	0.059

Abbreviations: AML, acute myeloid leukemia; CR1, first complete remission; CR2, second complete remission; HSCT, hematopoietic stem cell transplantation; *n*, number; WBC, white blood cells.

**Table 2 cancers-13-00336-t002:** Impact of *KIT* mutations on kinetics of *RUNX1–RUNX1T1* transcript levels *.

Variables	*n* Unmutated vs. Mutated	Log_10_ Transformed Transcript Levels			Log Reduction		
		Unmutated	Mutated	*p*	Unmutated	Mutated	*p*
*KIT* mutations							
Diagnosis	96 vs. 70	0.68 ± 0.02	0.64 ± 0.03	0.295	-	-	-
Post-induction	87 vs. 62	−2.39 ± 0.11	−2.02 ± 0.11	0.021	−3.07 ± 0.10	−2.65 ± 0.12	0.009
Pre-HSCT	96 vs. 70	−4.02 ± 0.14	−3.32 ± 0.17	0.002	−4.70 ± 0.14	−3.95 ± 0.17	0.001
1 month after HSCT	43 vs. 40	−4.69 ± 0.21	−4.05 ± 0.24	0.044	−5.38 ± 0.21	−4.65 ± 0.24	0.024
3 months after HSCT	57 vs. 45	−5.07 ± 0.15	−4.63 ± 0.26	0.140	−5.78 ± 0.15	−5.24 ± 0.26	0.076
D816V *KIT* mutation							
Diagnosis	137 vs. 29	0.67 ± 0.02	0.61 ± 0.04	0.160	-	-	-
Post-induction	122 vs. 27	−2.30 ± 0.09	−1.92 ± 0.17	0.068	−2.97 ± 0.09	−2.52 ± 0.17	0.026
Pre-HSCT	137 vs. 29	−3.82 ± 0.12	−3.26 ± 0.25	0.055	−4.50 ± 0.12	−3.86 ± 0.25	0.031
1 month after HSCT	67 vs. 16	−4.55 ± 0.17	−3.68 ± 0.38	0.032	−5.21 ± 0.17	−4.25 ± 0.41	0.019
3 months after HSCT	85 vs. 17	−5.00 ± 0.14	−4.25 ± 0.44	0.049	−5.68 ± 0.14	−4.85 ± 0.46	0.029

Abbreviations: HSCT, hematopoietic stem cell transplantation; *n*, number; * *RUNX1–RUNX1T1* transcript levels were normalized with respect to the number of *ABL1* transcripts and expressed as copy numbers per 10^5^ copies of *ABL1*. Data were expressed as mean ± SEM.

**Table 3 cancers-13-00336-t003:** Multivariate analyses to evaluate the prognostic independency of D816V *KIT* mutation and *RUNX1–RUNX1T1* MRD status at each time point for predicting post-transplant relapse.

Model #1	Pre-HSCT		1 Month after HSCT		3 Months after HSCT	
	HR (95% CI)	*p* Value	HR (95% CI)	*p* Value	HR (95% CI)	*p* Value
*RUNX1–RUNX1T1* levels						
≥3 log reduction	1		1		1	
<3 log reduction	5.31 (2.06–13.65)	0.001	7.15 (1.59–32.11)	0.010	22.23 (5.03–98.23)	<0.001
*KIT* mutations						
Unmutated	1		1		1	
Mutated	2.14 (0.81–5.62)	0.123	4.07 (0.73–22.72)	0.110	2.69 (0.42–17.46)	0.299
Disease state						
CR1	1		1		1	
CR2	7.55 (2.0–28.47)	0.003	13.3 (2.18–81.41)	0.005	9.37 (1.31–66.87)	0.026
**Model #2**	**Relapse**		**1 Month after HSCT**		**3 Months after HSCT**	
	**HR (95% CI)**	***p* Value**	**HR (95% CI)**	***p* Value**	**HR (95% CI)**	***p* Value**
*RUNX1–RUNX1T1* levels						
≥3 log reduction	1		1		1	
<3 log reduction	4.89 (1.91–12.49)	0.001	4.76 (0.95–23.86)	0.058	20.50 (4.68–89.81)	<0.001
D816V *KIT* mutation						
Unmutated	1		1		1	
Mutated	4.10 (1.63–10.30)	0.003	4.56 (1.02–20.33)	0.047	4.33 (1.01–18.56)	0.049
Disease state						
CR1	1		1		1	
CR2	6.75 (1.86–24.5)	0.004	9.22 (2.04–41.66)	0.004	7.59 (1.31–44.20)	0.024

Abbreviations: Allo-HSCT, allogeneic HSCT; Auto-HSCT, autologous HSCT; CI, confidence interval; CR1, first complete remission; CR2, second complete remission; HR, hazard ratio; HSCT, hematopoietic stem cell transplantation; *n*, number.

## Data Availability

The data presented in this study are available on request from the corresponding author. The data are not publicly available due to ethical considerations.

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
