# Peer review of "Prognostic Impacts of D816V KIT Mutation and Peri-Transplant RUNX1–RUNX1T1 MRD Monitoring on Acute Myeloid Leukemia with RUNX1–RUNX1T1"

_cancers, 2021, doi:10.3390/cancers13020336_

Round 1

Reviewer 1 Report

This revised paper was not significantly improved. The overall quality of paper is still not very high. For example, it is difficult to understand what the authors suggested treatment is for the pre-transplant MRD positive patients with D816V KIT mutation in the sentence of "Very poor outcomes of pre-transplant MRD positive patients with D816V KIT mutation suggested that this group should be treated in clinical trials" (Lines 50 – 51, page 2).

This study was designed to explore prognostic impacts of D816V KIT mutation and pre-transplant RUNX1-RUNX1T1 MRD monitoring on acute myeloid leukemia with RUNX1-RUNX1T1. Although the total sample size of 166 patients is not small, however, 74.1% patients are MRD- & D816V KIT- (123 subjects). Only 7 MRD+ & D816V KIT+ patients were observed in this study (6 patients underwent Allo-HSCT and 1 patient underwent Auto-HSCT).

English language should be globally revised. For example, one of double "in" in the sentence of "Auto-HSCT was superior to Allo-HSCT in in MRD negative patients without D816V KIT mutation (lines 48 – 49, page 2)" should be deleted.

Reviewer 2 Report

The edits made by the authors are fine by me and I do not have additional comments. Thank you.

Author Response

We really appreciated your positive feedback.

Reviewer 3 Report

the paper is very interesting but the authors did not add the required information at the first review. In particular the role of GvHD should be considered for patients undergoing allogeneic HSCT with positive MRD.It is necessary to add this time dependent covariate for OS and CIR endpoints.

The occurrence of GvHD and its treatment influence these important endpoints and must be considered in the analysis.

Round 2

Reviewer 3 Report

Authors have fully replied to all criticism,in particular the role of acute and chronic GvHD has been elucidated.

This manuscript is a resubmission of an earlier submission. The following is a list of the peer review reports and author responses from that submission.

Round 1

Reviewer 1 Report

This is an interesting retroperspective study of the prognostic impact of KIT D816V mutations and MRD after allogenic or autologous HSC transplantation in AML. The paper is well-written and the statistical analyses are sound.

Reviewer 2 Report

This study was designed to explore prognostic impacts of D816V KIT mutation and pre-transplant RUNX1-RUNX1T1 MRD monitoring on acute myeloid leukemia with RUNX1-RUNX1T1. The topic is interesting, however, this paper was not well written, the data was not well presented, , and sample size is not large. The overall quality of paper was not very high. This study does not significantly advance the knowledge in this field.

The followings are a few examples:
The spelling should be thoroughly examined. For example, "peri-transplant" in title should be "pre-transplant".

Introduction section should be reorganized, concisely providing brief background and the objective of this study.

Results section should be thoroughly revised. For example, it is hard to understand what does "96 vs. 70" in column "n" in Table 2 mean.

Reviewer 3 Report

The manuscript is an interesting and detailed analysis showing that in the hematopoietic stem cell transplantation setting of high-risk Acute Myeloid Leukemia (AML) patients the detection of transcript levels of the RUNX1-RUNX1T1 fusion transcripts and D816V KIT mutations is able predict relapse and disease outcome. The article is well written, with a good structure, the figures and tables are clear and informative making the manuscript easy to follow and understand. The overall message of the manuscript is clear and concise and the conclusions are supported by the data illustrated in the figures and tables.

Nevertheless its significance and value for the scientific and medical community these issues should be addressed:

  • In line 94 it should be “performed” instead of “preformed”.
  • A short query on Google retrieved one article that is not cited in the manuscript and the authors should include and incorporate it in the text: Krauth et al, 2014 (DOI: 10.1038/leu.2014.4. Epub 2014 Jan 9).
  • The phrase in lines 145-148 should be clarified and explained better.
  • In Table 3 are shown Multivariate analyses using two different Models (#1 and #2) but none is clearly explained in the text what it mean exactly.
  • Self-citation: this manuscript uses 37 references and at least 9 of which are references from the authors themselves. It is expected a certain degree of self-citation (according to the latest studies point to an average of 12.5%) but in this article it reaches 24.32% which is almost the double. This should be revised accordingly.

Reviewer 4 Report

The authors demonstrate that mutated CBF AMLs which have favorable MRD kinetics pre and post transplant have a better outcome. The data is a single center retrospective analysis. Study design There are some challenges in methodology in this study - particularly as high dose ARA-C consolidation was not used- this would be standard of care in many parts of the world. Also, there are many publications which have shown an advantage for Gemtuzumab in CBF AML therapy. Moreover, auto-transplant would not necessarily be standard of care. There is no discussion around no use of high dose AraC or GO in induction or consolidation . These flaws must be clearly addressed in the discussion in the context of the changing era of therapy for CBF AML as these treatments have not been used in this study. Methods The entire data depends on the reproducibility and accuracy of the MRD assessments. There needs to a section in the methods to cover this important area. How as the external QC for MRD assays established. Were there interlab comparisons with a reference laboratory? What were the limits of detection and how was reproducibility for the assay established? Was the 3 log reduction confirmed on a second sample? Results Graphs for KP curves are too small to read and must be redone. Fig 1. They are also confusing to read. Perhaps best presented as KIT mutated vs unmutated or the specific mutation D816V and allo and auto HSCT clearly separated. Fig.3. These can be reduced to 3 or 4 key graphs. Non relapse mortality can be supplied as an appendix figure. Fig 4. Present negative results in appendix figure

Reviewer 5 Report

This study retrospectively analyzed a cohort of high-risk AML patients (n=166) with RUNX1-RUNX1T1 fusion and demonstrated the differentiated prognostic impact of different KIT mutations with KIT D816V being associated with post-transplant relapse and poor prognosis.  The role of Allo- vs Auto HSCT was depicted with auto-HSCT that was superior to Allo-HSCT in in MRD negative patients without D816V KIT mutation. In contrast, Allo-HSCT was superior to Auto-HSCT in MRD negative patients with D816V KIT mutation.

Major comments:

  • Did the authors analyzed additional genetic alterations (e.g. RAS mutations or mutations in other genes) that can be correlated with relapse?
  • Please provide a multiple testing correction for the p values in table 1.
  • The authors should provide a biological interpretation on the different prognostic impact of KIT mutations and discuss it.

Reviewer 6 Report

The text by Byung -Sik Cho et coll. is very interesting because it evaluates in a combined way the role of 2 factors in AML patients: RUNX1-RUNX1T1 and KIT mutations in adult patients undergoing allogeneic or autologous stem cell transplantation. Sections of materials and methods, results and conclusions are presented in a clearly readable manner also for readers not specifically involved in the field of hematopoietic stem cell transplantation. Also the references section is complete.

Issues:

1) the time frame in which the transplants were performed is not defined: being a large number of cases I believe that this data should be included

2) the definition of a) myeloablative regimen, b) reduced intensity regimen, c) non-myeloablative regimen is missing 3) the authors report the use of ATG at of 5 mg/kg in the case of haploidentical transplantation, while in the case of HSCT from an unrelated or even family donor it is not mentioned. In particular, the concept of omitting the ATG in the case of HSCT from an unrelated donor is to be stressed. In Europe the standard of practice for MUD transplants includes ATG especially for most frail patients

4) the text does not report the appearance/role of GvHD as a post-transplant complication but also the role in controlling the disease. The incidence of acute and chronic GvHD must be described and included among the covariates both for the analysis of the risk of relapse/TRM and survival. In particular, this analysis is very important as MSD and MUD patients were transplanted without ATG

5) it would be interesting to see the outcome results according to the type of donor MSD, MUD 10/10 and 9/10 or 8/10 and haploidentical, in other words the authors can highlight a protective role towards relapse depending on the type of donor used?

6) considering the large number of patients treated uniformly, the authors can indicate the distribution of KIT mutations as a function of age (eg 18-30, 31-40 etc).
